# A Novel Score Using Lymphocyte-to-Monocyte Ratio in Blood and Malignant Body Fluid for Predicting Prognosis of Patients with Advanced Ovarian Cancer

**DOI:** 10.3390/cancers15082328

**Published:** 2023-04-17

**Authors:** Min Jin Jeong, Yeo Nyeong Yoon, Yeon Kyung Kang, Chan Joo Kim, Hae Seong Nam, Yong Seok Lee

**Affiliations:** 1Department of Obstetrics and Gynecology, Eunpyeong St. Mary’s Hospital, College of Medicine, The Catholic University of Korea, Seoul 03312, Republic of Korea; mjobgy@catholic.ac.kr (M.J.J.);; 2Department of Obstetrics and Gynecology, Uijeongbu St. Mary’s Hospital, College of Medicine, The Catholic University of Korea, Seoul 11765, Republic of Korea; 3Division of Pulmonology, Department of Internal Medicine, Inha University Hospital, Inha University School of Medicine, Incheon 22332, Republic of Korea

**Keywords:** advanced ovarian cancer, malignant body fluid, systemic inflammation, lymphocyte-to-monocyte ratio, prognostic factor

## Abstract

**Simple Summary:**

In recent decades, research efforts have enabled an increased awareness of critical roles of systemic inflammatory markers in malignant body fluids of ovarian cancer patients. The aim of this study was to investigate the prognostic value of the lymphocyte-to-monocyte ratio (LMR) in malignant body fluid (malignant pleural effusion or malignant ascites) (mLMR) and the combination of LMR in peripheral blood (bLMR) and mLMR for predicting the prognosis of patients with advanced-stage epithelial ovarian cancer. The results highlighted that both low bLMR (*p* = 0.013) and low mLMR (*p* = 0.006) were associated with poor prognosis of ovarian cancer. Furthermore, combined scoring of these two factors (*p* < 0.001) was found to be a better factor for predicting disease recurrence of ovarian cancer.

**Abstract:**

(1) Background: The lymphocyte-to-monocyte ratio (LMR), one of the systemic inflammatory markers, has been shown to be associated with prognosis of various solid tumors. However, no study has reported clinical utility of the LMR of malignant body fluid (mLMR) (2) Methods: We retrospectively analyzed clinical data of the final 92 patients of a total of 197 patients with advanced ovarian cancer newly diagnosed from November 2015 and December 2021 using our institute big data. (3) Results: Patients were divided into three groups according to their combined bLMR and mLMR scores (bmLMR score): 2, both bLMR and mLMR were elevated; 1, bLMR or mLMR was elevated; and 0, neither bLMR nor mLMR was elevated. A multivariable analysis confirmed that the histologic grade (*p* = 0.001), status of residual disease (*p* < 0.001), and bmLMR score (*p* < 0.001) were independent predictors of disease progression. A low combined value of bLMR and mLMR was strongly associated with a poor prognosis in patients with ovarian cancer. (4) Conclusions: Although further studies are required to apply our results clinically, this is the first study to validate the clinical value of mLMR for predicting prognosis of patients with advanced ovarian cancer.

## 1. Introduction

Ovarian cancer is the eighth most common cancer in women worldwide. Based on cancer statics, 313,959 new cancer cases and 207,252 cancer deaths are recorded in women annually [1]. The overall 5-year relative survival rate of patients with ovarian cancer is less than 50% over three decades [2]. Compared to patients with other gynecological malignancies, patients with ovarian cancer suffer from the highest mortality rate. For these reasons, there is a major unmet need for a variety of new biomarkers to determine effective treatment and predict the prognosis in patients with ovarian cancer.

Accumulating evidence has shown that inflammation plays a critical role in tumor development and progression. It is one of the hallmarks of cancer [3]. This scientific validity has been stimulated and expanded by research on various cancer-associated inflammatory biomarkers. Among different biomarkers, the neutrophil-to-lymphocyte ratio of blood (bNLR) and the lymphocyte-to-monocyte ratio of blood (bLMR) that could be easily and inexpensively measured in clinical practice, have been examined over the past decade to predict the response of treatment and the survival of patients with various cancers, including ovarian cancer [4,5,6,7,8].

Approximately 75% of patients with epithelial ovarian cancer are diagnosed at advanced stages (III-IV), which include tumor spread into the pleural space or peritoneal cavity [9]. Malignant pleural effusions in ovarian cancer most probably result from pleural invasion from contiguous structures (such as the diaphragm) or transdiaphragmatic migration of malignant cells thorough pleuroperitoneal communications [10]. Malignant ascites are often indicative of peritoneal carcinomatosis or the presence of malignant cells in the peritoneal cavity [11]. These malignant body fluids constitute a specific microenvironment reflecting carcinogenesis. Thus, they may influence treatment strategies and disease progression. Recently, incredible advancement in liquid biopsy has shown the potential that body fluids such as urine, pleural effusion, ascites, and bronchial samples can be better repository of biomarkers of tumor origin than blood [12]. Starting from this validity, it is not surprising that the NLR not only in blood but also body fluids can predict prognosis [13,14].

Similarly, low bLMR has been associated with a poor prognosis in various cancers including ovarian cancer. Several studies indicated that a low bLMR value was correlated with poor prognosis in patients with resectable colorectal and pancreatic head cancer. Moreover, pretreatment bLMR was found to be an independent prognostic factor for patients with head and neck cancer treated by concurrent chemoradiotherapy [4,7,15,16,17]. However, no study on the LMR of malignant body fluid (mLMR), which is present in more than half of patients with advanced ovarian cancer, has been reported thus far. With this in mind, we questioned whether mLMR in ovarian cancer patients might have a clinical impact similar to NLR. Accordingly, we evaluated values of mLMR from ovarian cancer patients and investigated the clinical benefit of using bLMR and mLMR for predicting patient outcomes.

## 2. Materials and Methods

### 2.1. Study Population and Data Collection

Clinical data of patients with epithelial ovarian cancer who underwent debulking operation or chemotherapy at the Department of Gynecologic Oncology, Seoul St. Mary’s Hospital and seven branch hospitals from November 2015 and December 2021 were collected. We used the Clinical Data Warehouse (CDW). The CDW integrates all eight hospitals affiliated with The Catholic Medical College of Korea. It is a research-supporting system with de-identified clinical information. Data contained information about 2.8 million patients, 47 million prescription events, and laboratory results for 150 million cases. All files of extracted data were encoded to prevent personal identification of patients during the sorting process.

From 2015–2021, blood and ascites/pleural fluid samples were collected from patients with newly diagnosed advanced (stage III or IV) ovarian cancer. Blood was collected within two weeks prior to primary debulking surgery or neoadjuvant chemotherapy. Pleural effusions were obtained by thoracentesis and ascites were collected by diagnostic paracentesis or in the operating room during surgery. All samples were collected into EDTA tubes and analyzed within two hours after collection using an automated hematology analyzer (Sysmex XN-350). Analyses of differential white blood count and malignant cells of body fluid were performed utilizing whole blood mode and body fluid mode, respectively [18]. In addition, malignant cell detection in body fluid was confirmed with microscopic method and cell block results. Medical records of patients were retrospectively reviewed for prognostic clinicopathological and laboratory variables, tumor stage and grade, baseline serum lactate dehydrogenase (LDH) levels and CA125 levels, status of residual disease, and neoadjuvant chemotherapy. Cancer stages were re-assessed for all patients according to FIGO (The International Federation of Gynecology and Obstetrics) 2018 staging system. Unstaged patients were excluded from this analysis if the patient gave up midway because she did not want to be treated or was lost to follow-up for personal reasons. Patients were also excluded if they had a history of another malignancy within the previous five years or other diseases associated with systemic inflammation, such as hematologic disorder, rheumatic disease, and connective tissue disorder. Additionally, patients who were taking nonsteroidal anti-inflammatory drugs, oral contraceptive drugs, or anticoagulant medications before sampling were excluded. Two patients with clinical signs of sepsis and one patient with a final diagnosis of Meig’s syndrome were also excluded from this study.

A total of 197 patients (128 with ascites and 69 with pleural effusions) were found in big data and 124 patients were selected after excluding those with cancer cells not found in the final cytology results. Of these, 92 people were finally enrolled in the study after excluding cases that met the exclusion criteria described earlier. The LMR was defined as the absolute lymphocyte count divided by the absolute monocyte count in the differential cell count of peripheral blood or malignant body fluid (ascite, pleural fluid). This study was approved by the Institutional Review Board of the Catholic Medical College of Korea (reference number: XC20WIDI0099). Informed consent was waived because of the retrospective nature of this study.

### 2.2. A New Score Using LMRs of Peripheral Blood and Malignant Body Fluid (bmLMR Score)

Since there was no defined mLMR value for patients with ovarian cancer, optimal cutoff values for LMRs of peripheral blood and malignant body fluid were determined using maximally selected rank statistics [19]. These were calculated using the maxstat package in R software, version 4.1.0 (R Foundation for Statistical Computing, Vienna, Austria). According to cutoff values for bLMR and mLMR, the bmLMR score was defined as follows as mentioned in our previous study [10]: patients in whom both bLMR (≥2.80) and mLMR (≥2.41) were elevated were assigned a score of 2. Patients in whom only one of these two LMR values was elevated were assigned a score of 1. Patients in whom neither bLMR nor mLMR values was elevated were assigned a score of 0.

### 2.3. Statistical Analysis

Progression-free survival (PFS) was defined as the time from the date of diagnosis of ovarian cancer to the date of the first observation of tumor recurrence or death. Progression was estimated based on imaging tests (computed tomography, magnetic resonance imaging, abdominal ultrasound, chest X-ray, or PET-CT) performed during the follow-up period.

Clinical and histopathological factors showing significant associations with prognosis in the univariate analysis were entered into a multivariate Cox regression model (forward sequential method) to determine their independent effects. Results of the Cox regression modeling are expressed as hazard ratios (HRs) and associated 95% confidence intervals. Parameters included in the multivariate analysis model were as follow: histologic subtype, grade, optimal debulking surgery, bLMR, bNLR, mLMR, mNLR, and bmLMR score. The Kaplan–Meier method was used for survival analysis, and the log-rank test was used to compare the differences between groups. A *p*-value equal to or less than 0.05 indicated statistical significance. All analyses were performed using SPSS statistical software package, version 23 (IBM SPSS Statistics for Windows, Version 23.0, Armonk, NY, USA).

## 3. Results

### 3.1. Baseline Characteristics

From 124 consecutive patients with malignant body fluid, only 92 patients were analyzed in this study, including 55 patients with malignant ascitic fluid and 37 patients with malignant pleural fluid. Of the 55 patients with malignant ascites, 44 were based on peritoneal washing cytology results performed at the time of surgery. The remaining 11 were based on cytology results through paracentesis or puncture drainage to control symptoms before treatment. All 37 cases with malignant pleural effusion underwent thoracentesis before treatment. There was no patient with simultaneous pleural effusion and ascites in our study. Baseline characteristics of the study population and tumors are summarized in Table 1. The median age at time of diagnosis was 58.2 years (range, 36–82 years). The majority (84.8%) of patients presented with a serous type, whereas 8.7%, 4.3%, and 2.2% of patients had a mucinous type, clear cell carcinoma, and other histologic types, respectively. All patients had FIGO stages III–IV with a histologic grade of high (n = 53) or moderate to mild (n = 39). The percentage of patients who had a normal body weight (BMI: 18.5–22.9 kg/m^2^) was 36.9% and that of patients who were overweight or obese (BMI ≥ 23 kg/m^2^) was 63.1%. The optimal debulking status was defined as follows: R0, no macroscopic residual disease; optimal, remaining disease 0.1–1 cm; and suboptimal, remaining visible disease > 1 cm. According to cutoff values for bLMR and mLMR, all patients were classified into one of three bmLMR score groups as follows: score of 0, 34 (35.8%) patients; score of 1, 42 (44.2%) patients; and score of 2, 16 (16.8%) patients.

At the time of diagnosis, 81.5% of patients were treated with a primary debulking surgery. Among them, 66.3% underwent R0 and optimal debulking surgeries. Seventeen patients received neoadjuvant chemotherapy before debulking surgery. The median follow-up time was 21.7 months (range, 2–144 months). The number of patients who recurred within the observation period was 75 (81.5%). A total of 69 (75%) patients died. Among the patients who died, 64 died of ovarian cancer, and 5 died for other reasons.

### 3.2. Univariate and Multivariate Analyses of Prognostic Factors for Progression-Free Survival

The median PFS of all patients was 29.2 months (95% CI: 20.3~30.8 months). Results of the univariate and multivariate analyses of clinicopathological parameters are summarized in Table 2. Univariate analysis revealed that PFS was significantly influenced by histologic grade (*p* < 0.001), histologic subtype (*p* = 0.044), and result of optimal debulking (*p* < 0.001). CA-125 and LDH failed to show statistically significant association with PFS. Systemic inflammatory markers such as the NLRs and LMRs (bNLR, mNLR, bLMR, mLMR, bmLMR score) were also significant prognostic factors in the univariate analysis. In the multivariate analysis, histological grade (adjusted HR: 2.40; 95% CI: 1.44–4.01; *p* = 0.001), optimal debulking surgery (adjusted HR: 0.34; 95% CI: 0.20–0.58; *p* < 0.001), and the bmLMR score (adjusted HR: 3.36; 95% CI: 0.67–6.75; *p* < 0.001) were independent predictor for PFS (Figure 1).

## 4. Discussion

In this present study, we found that inflammation related factors, not only low bLMR, but also low mLMR, were associated with poor prognosis of ovarian cancer. Furthermore, the bmLMR score was found to be a better factor for predicting disease recurrence of ovarian cancer. To the best of our knowledge, there are no reports describing the prognostic relevance of mLMR in ovarian cancer.

Advanced ovarian cancer patients with old age, high tumor grade, histopathologic subtype, status of residual disease after operation and presence of ascites have been found to have worse prognosis [20,21]. In addition to such already known factor, increasing data have shown that peripheral blood cells and relevant ratios are related to tumor prognosis, such as the NLR and LMR [15,16,17]. In particular, absolute lymphocyte and monocyte counts have been demonstrated to be associated with prognosis in ovarian cancer [4,5,6,7,8]. The precise mechanism by which low LMR indicates poor outcome remains unclear. However, previous studies have suggested that a relatively lower number of lymphocytes and an excess of monocytes might play a crucial role in tumor microenvironment [5,6,15,16,17]. Lymphocytes play critical roles in host immune responses. They show potent anticancer activities that can inhibit growth and metastasis of several tumors [3]. In particular, tumor-infiltrating lymphocytes can induce cytotoxic cell death and inhibit tumor cell proliferation and migration by regulating immune interactions [22]. According to this mechanism, a low lymphocyte count indicates a weakened anti-tumor immune response, which may predict a poor prognosis [23]. Monocytes are also associated with tumorigenesis. They can suppress host anticancer immune responses [3]. In addition, cytokines and chemokines produced by tumor cells can trigger differentiation of monocytes into tumor-associated macrophages (TAM) [24]. TAM can promote tumor cell invasion and tumor-related angiogenesis [24]. Therefore, increased peripheral monocytes in the tumor microenvironment are closely related to tumor progression [25,26].

Numerous studies have shown that a low LMR predicts a shorter PFS, whereas a high LMR indicates a longer PFS in ovarian cancer [4,7,8,27]. Zhu et al. performed a large cohort multicenter study about peripheral blood LMR in 672 patients with advanced epithelial ovarian cancer [7] and found that a low LMR is associated with a poor response to neoadjuvant chemotherapy and a worse disease progression. Eo et al. also retrospectively examined bLMR as a prognosticator in a cohort of 234 patients with epithelial ovarian cancer who have undergone a debulking surgery [8]. Their data showed that low LMR was significantly correlated with a poor disease progression. In our study, univariate analysis revealed that PFS was significantly influenced by bLMR and mLMR. Furthermore, multivariable analysis confirmed that only the bmLMR score was an independent predictor of disease progression.

Recently, it has been shown that ascites-associated ovarian cancer may provide an immuno-suppressive environment for disease development [28]. Like peripheral blood, malignant body fluid also contains tumor cells and immune cells including lymphocytes, macrophages, and dendritic/NK cells. Thus, it might also provide a local tumor microenvironment [29,30]. We have previously performed a retrospective study about the prognostic impact of inflammatory markers in malignant pleural effusion of lung cancer patients [14]. We found that the NLR in peripheral blood and MPE were significant prognostic factors for adverse overall survival. In ovarian cancer, however, a paper that analyzes specific systemic inflammatory marker as a prognostic factor in malignant pleural effusion or ascites has not been reported. Therefore, we are reporting this for the first time. Nevertheless, malignant body fluid in ovarian cancer is already showing promise both as a predictor of disease progression and as a substrate for monitoring drug response in immune target therapy [31,32,33]. Lane et al. studied inflammation-regulating factors in ascites with advanced serous epithelial ovarian carcinoma [31] and found that IL-6 has a significant association with PFS in multivariate analysis. Chen et al. revealed that targeted cytokine (INF-gamma) expression level in ascites could serve as an immune biomarker for predicting outcomes in ovarian cancer [32]. Moreover, in a study by Zhang and colleagues, a high level of senescent T cells in ascites showed positive correlations with a short PFS and chemoresistance in advanced high grade serous ovarian cancer [33]. Of course, the generation of cell-mediated immune response depends on interactions of various cytokines. Focusing on inflammatory cell themselves rather than on each cytokine could directly reflect the whole inflammatory process.

A few research studies have suggested a new scoring system by combining inflammatory hematologic markers as an independent prognostic factor for PFS [10,27]. For example, Tang et al. performed a retrospective study on combined preoperative peripheral blood LMR and CA125 in ovarian cancer patients [27] and found that a low LMR and a high CA-125 group have significant associations with poor PFS in multivariate analysis. However, the specificity of CA125 as a prognostic factor is already known to be relatively low. In addition, CA125 could not reflect inflammatory response, unlike the LMR [34]. Thus, we tried to incorporate bmLMR into our study by combining bLMR and mLMR and investigated their correlations with recurrence. Consequently, our novel combined bmLMR score was a more valuable prognostic factor than bLMR or mLMR alone for recurrence of ovarian cancer.

In real clinical practice, physicians often consider the need for additional, aggressive treatment. Therefore, if we can accurately predict a patient’s prognosis in a less invasive and cheaper way, it can be of enormous help in patient care. Based on recent research, gynecologic oncologists consider maintenance treatment strategies such as poly (ADP-ribose) polymerase (PARP) inhibitors and antiangiogenic agents more positively after archiving a complete response with standard platinum-based front-line chemotherapy for patients with newly diagnosed advanced ovarian cancer. However, despite these proven benefits of maintenance therapy, the toxicity and expensive cost of the drugs should also be considered. At the point of hesitation whether to start maintenance treatment, useful and cheap clinical biomarkers that could predict disease recurrence will benefit clinicians in determining the maintenance therapy for patients with advanced ovarian cancer. Although further validation studies of relationship between these parameters and BRCA and using larger cohorts are warranted to generalize our findings, the bmLMR scoring has potential to help in this clinical dilemma for clinicians.

In addition, the LMR can be readily calculated at a low-cost and LMR is universally available in clinical settings from different cell counts using peripheral blood and body fluid. In patients with advanced ovarian cancer, malignant body fluid drainage is often performed for diagnosis or symptom relief before surgery and body fluid cytology are mandatory for tumor staging during debulking operation. Therefore, it is possible to easily obtain a sample and confirm the bmLMR scoring, which has the advantage of high utilization in clinical practice. Thus, we believe that the bmLMR score could potentially be an attractive and ideal biomarker that might provide valuable additional prognostic information in advanced ovarian cancer.

The present study had several limitations. First, our study was not validated in an independent series of patients with a relatively small sample size, which limits our ability to generalize the findings. Thus, a study with a larger sample is needed to evaluate our results in different clinical settings and to establish a reliable cutoff value for the LMR in peripheral blood and body fluid. Second, all data were retrospectively collected using electronic medical records. Thus, clinical and recurrence comparison might have been influenced by selection bias due to its retrospective nature. Lastly, the LMR counts in peripheral blood and body fluid depend on a wide range of factors such as acute or chronic infection, inflammatory disease, and personal lifestyle habits. We considered each patient’s medical condition and medications through the data obtained. Moreover, body fluid was obtained by inconsistent methods such as puncture drainage, paracentesis/thoracentesis, debulking operation (peritoneal washing cytology), and different physicians (gynecologic oncologist or radiologist in multi-medical institutions). Such differences might influence the ratio and the score due to dilution or different counts. Accordingly, it was difficult to elucidate other factors in this retrospective analysis.

However, to maintain the quality of the present study, we included only patients in whom malignant cells were identified in the malignant body fluid. All patients were excluded if they had a history of another malignancy or other diseases associated with systemic inflammation. Additionally, patients taking nonsteroidal anti-inflammatory drugs (NSAIDs), oral contraceptive drugs, or anticoagulant medications before sampling were excluded.

## 5. Conclusions

Malignant body fluids and the associated tumor-promoting microenvironment can alter disease prognosis and therapy responses. Our results demonstrate that the bmLMR score at pretreatment is strongly associated with recurrence in patients with newly diagnosed advanced epithelial ovarian cancer. Although further studies are needed to clarify the exact biological mechanism and confirm our findings, the findings of this study will benefit clinicians and patients in determining the most appropriate management for patients with malignant body fluid. Finally, our data could serve as a basis for future research about the roles of systemic inflammatory markers in prognosis, the treatment responses in recurrent settings, and the effect of target immune therapy.

## Figures and Tables

**Figure 1 cancers-15-02328-f001:**
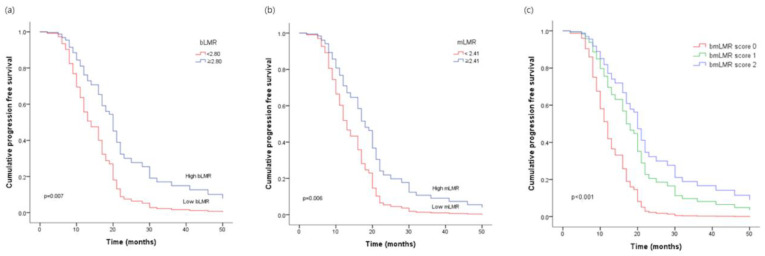
Progression-free survival of ovarian cancer patients according to (**a**) the lymphocyte-to-monocyte ratio in peripheral blood (bLMR), (**b**) the lymphocyte-to-monocyte ratio in malignant fluid (mLMR), (**c**) the new score: bmLMR score.

**Table 1 cancers-15-02328-t001:** Baseline patient’s characteristics (n = 92).

Clinical Characteristic	Measure, n (%)	
Age (years)	≥65	27 (29.3) ^1^
	<60	65 (70.7)
BMI (kg/m^2^)	Normal 18.5–22.9	34 (36.9)
	Overweight ≥ 23	58 (63.1)
Hypertension	Yes	31 (33.7)
	No	61 (66.3)
Diabetes	Yes	11 (11.9)
	No	81 (88.1)
Histologic type	Serous	78 (84.8)
	Non-serous	14 (15.2)
Stage (FIGO)	III	50 (54.3)
	IV	42 (45.7)
Histological grade	G1	2 (2.2)
	G2	37 (40.2)
	G3	53 (57.6)
Optimal debulking	Yes	61 (66.3)
	No	31 (33.7)
Neoadjuvant CTx	Yes	17 (18.5)
	No	75 (81.5)
Body fluid	Ascitic fluid	55 (59.8)
	Pleural fluid	37 (40.2)
mNLR	High ≥ 0.03	76 (82.6)
	Low < 0.03	16 (17.4)
bNLR	High ≥ 5.90	22 (23.9)
	Low < 5.90	70 (76.1)
mLMR	High ≥ 2.41	47 (51.1)
	Low < 2.41	45 (48.9)
bLMR	High ≥ 2.80	26 (28.3)
	Low < 2.80	66 (71.7)
bmLMR score	0	34 (35.8)
	1	42 (44.2)
	2	16 (16.8)
State	Survival	23 (25)
	Died	69 (75)
Recurrence	Yes	75 (81.5)
	No	17 (18.5)
CA-125 (U/mL) ^2^	High	89 (96.7)
	Normal	3 (96.7)
LDH (U/L) ^2^	High	84 (97.7)
	Normal	2 (2.3)

^1^ Data in parentheses are percentages. ^2^ CA-125, LDH: Dichotomized by cutoff of normal value. Abbreviations: BMI, body mass index; FIGO, International Federation of Gynecology and Obstetrics; mNLR, neutrophil-to-lymphocyte ratio of malignant body fluid; bNLR, neutrophil-to-lymphocyte ratio of peripheral blood, mLMR, lymphocyte-to-monocyte ratio of malignant body fluid; bLMR, lymphocyte-to-monocyte ratio of peripheral blood; LDH, lactate dehydrogenase.

**Table 2 cancers-15-02328-t002:** Univariate and multivariate analysis of prognostic factors for the progression-free survival (n = 92).

	Univariate		Multivariate	
	HR (95% Cl)	*p*-Value	Adjusted HR (95% Cl)	*p*-Value
Age (years)		0.371		
≥65	1.26 (0.75–2.13) ^1^			
<65	1.00 (Ref.)			
BMI		0.034		
normal =< 23	0.76 (0.60–0.98)			
overweight > 23	1.00 (Ref.)			
Histological grade		<0.001		0.001
G3	1.58 (1.24–2.01)		2.40 (1.44–4.01)	
G1/G2	1.00 (Ref.)		1.00 (Ref.)	
Optimal debulking		<0.001		<0.001
yes	0.55 (0.42–0.71)		0.34 (0.20–0.58)	
no	1.00 (Ref.)		1.00 (Ref.)	
Histological subtype		0.044		
serous	2.06 (1.01–4.17)			
others	1.00 (Ref.)			
Neoadjuvant CTx.		0.596		
yes	1.16 (0.66–2.05)			
no	1.00 (Ref.)			
bLMR		0.013		
≥2.80	0.51 (0.30–0.87)			
<2.80	1.00 (Ref.)			
bNLR		0.008		
≥5.90	2.03 (1.20–3.45)			
<5.90	1.00 (Ref.)			
mLMR		0.006		
≥2.41	0.52 (0.33–0.83)			
<2.41	1.00 (Ref.)			
mNLR		0.012		
≥0.03	2.46 (1.22–4.98)			
<0.03	1.00 (Ref.)			
LDH (U/L)		0.549		
≥271	1.00 (Ref.)			
<271	0.64 (0.15–2.67)			
CA-125 (U/mL)		0.386		
≥35	1.86 (0.45–7.62)			
<35	1.00 (Ref.)			
CRP (mg/dL)		0.952		
≥0.5	1.02 (0.40–2.59)			
<0.5	1.00 (Ref.)			
albumin (g/dL)		0.897		
≥3.5	1.00 (Ref.)			
<3.5	1.03 (0.63–1.68)			
bmLMR score		0.001		<0.001
0	1.00 (Ref.)		1.00 (Ref.)	
1	0.32 (0.16–1.64)		1.40 (0.71–2.75)	
2	0.43 (0.26–0.72)		3.36 (0.67–6.75)	

^1^ Data in parentheses are percentages. Abbreviations: HR, hazard ratio; CI, confidence interval.

## Data Availability

The data presented in this study are available on request from the corresponding author.

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
