# Peer review of "A Novel Score Using Lymphocyte-to-Monocyte Ratio in Blood and Malignant Body Fluid for Predicting Prognosis of Patients with Advanced Ovarian Cancer"

_cancers, 2023, doi:10.3390/cancers15082328_

Round 1

Reviewer 1 Report

A well written, original and interesting manuscript

Comment 1: In the statistical section do you mean death from any reason? Due to recurrence only? Please specify.

Comment 2: Were the PFS curves evaluated by Kaplan Meier method with a log rank test? Please add in the statistical section

Comment 3: pages 8-9 lines 272-280. This part of the discussion is not based on your data because you did not correlate the results with BRCA status of chemotherapy response score. Please rephrase. It can be said that in further larger studies relationship to these parameters can be done to help treatment decisions…

Comment 4: in the manuscript limitations it should include the fact that the analyzed group was not homogenous. The fluid was obtained from ascites or pleural effusion and also some of the abdominal specimens were from washings while others from ascites. Presence of ascites does signify worse disease spread and prognosis. Thus such differences might influence the ratio and the score due to dilution or different counts  

Reviewer 2 Report

The current study constructed a prognosis prediction model on the survival of advanced ovarian cancer. I have the following comments.

·      Validation of the model here by an external dataset is suggested. If not possible, please discuss this as a limitation.

·      lymphocyte to monocyte ratio” is commonly adopted in cancer research, please discuss the role of this biomarker in other cancer types.

·      Background information about ovarian cancer survival is sugged to include in the introduction section (PMID: 30310512).
